# Soil Fungal Community Structure and Its Effect on CO_2_ Emissions in the Yellow River Delta

**DOI:** 10.3390/ijerph20054190

**Published:** 2023-02-26

**Authors:** Linhui Ji, Yu Xin, Dufa Guo

**Affiliations:** College of Geography and Environment, Shandong Normal University, Jinan 250014, China

**Keywords:** CO_2_ emissions, fungal community, structure characteristics, adaptation mechanisms, different salinity gradients

## Abstract

Soil salinization is one of the most compelling environmental problems on a global scale. Fungi play a crucial role in promoting plant growth, enhancing salt tolerance, and inducing disease resistance. Moreover, microorganisms decompose organic matter to release carbon dioxide, and soil fungi also use plant carbon as a nutrient and participate in the soil carbon cycle. Therefore, we used high-throughput sequencing technology to explore the characteristics of the structures of soil fungal communities under different salinity gradients and whether the fungal communities influence CO_2_ emissions in the Yellow River Delta; we then combined this with molecular ecological networks to reveal the mechanisms by which fungi adapt to salt stress. In the Yellow River Delta, a total of 192 fungal genera belonging to eight phyla were identified, with Ascomycota dominating the fungal community. Soil salinity was the dominant factor affecting the number of OTUs, Chao1 index, and ACE index of the fungal communities, with correlation coefficients of −0.66, 0.61, and −0.60, respectively (*p* < 0.05). Moreover, the fungal richness indices (Chao1 and ACE) and OTUs increased with the increase in soil salinity. *Chaetomium*, *Fusarium*, *Mortierella*, *Alternaria*, and *Malassezia* were the dominant fungal groups, leading to the differences in the structures of fungal communities under different salinity gradients. Electrical conductivity, temperature, available phosphorus, available nitrogen, total nitrogen, and clay had a significant impact on the fungal community structure (*p* < 0.05). Electrical conductivity had the greatest influence and was the dominant factor that led to the difference in the distribution patterns of fungal communities under different salinity gradients (*p* < 0.05). The node quantity, edge quantity, and modularity coefficients of the networks increased with the salinity gradient. The Ascomycota occupied an important position in the saline soil environment and played a key role in maintaining the stability of the fungal community. Soil salinity decreases soil fungal diversity (estimate: −0.58, *p* < 0.05), and soil environmental factors also affect CO_2_ emissions by influencing fungal communities. These results highlight soil salinity as a key environmental factor influencing fungal communities. Furthermore, the significant role of fungi in influencing CO_2_ cycling in the Yellow River Delta, especially in the environmental context of salinization, should be further investigated in the future.

## 1. Introduction

Saline soil accounts for more than 7% of the earth’s land surface, and approximately 70% of saline soil is used for agricultural production [1]. However, the amount of saline soil increases by 10% every year [2], which is one of the most compelling environmental problems on a global scale. High salinity in soil inhibits plant growth, changes the soil’s physical and chemical properties, and even causes the degradation of soil quality [3]. Fungi are the main members of soil microorganisms and are widely distributed in terrestrial ecosystems. Fungi play a key role in organic matter decomposition [4] and nutrient cycling [5] in the soil ecosystem, and the fungal community structure is often used as an important parameter for measuring the change in soil quality [6]. Understanding the response of the soil fungal community to the change in salinity is essential for rehabilitating salinized soil. Previous studies have reported the effects of salinity on soil microbial biomass [7], microbial activity [8], diversity [9,10], community composition [11,12], soil enzyme activity [13,14], and soil physical and chemical properties [15]. However, studies regarding the structural fluctuation and interactions with the soil fungal community among the natural salinity gradient during the process of regional evolution remain scarce. Moreover, soil fungi are active participants in the soil carbon and nutrient cycles; organic matter is decomposed, altered, and modified by soil fungi. The products of these processes are typically greenhouse gases such as CO_2_ that are released into the atmosphere [16]. However, plants can form symbiotic relationships with mycorrhizal fungi that firmly anchor carbon in the soil, and scientists have learned that a special type of mycorrhizal fungus, i.e., ectomycorrhizal fungi, is helping plants to take up carbon dioxide more quickly [17]. Therefore, the relationship between soil fungal communities and CO_2_ emission fluxes needs to be dissected to provide a fungal perspective for the future development of efficient microbial management strategies to mitigate greenhouse gas emissions [18].

In recent years, many scholars have been studying the changing characteristics of microbial salt stress in different regions. For example, Rath et al. assessed the microbial communities along two naturally occurring salt gradients located around Lake O’Connor in Western Australia and found that estimates of fungi were less affected by salinity than bacteria [19]. Zheng et al. evaluated soil prokaryotic microbial communities in Bohai Bay located in China and found that salinity altered the composition and structure of the prokaryotic microbial communities and enhanced their interactions [20]. This shows that the effect of salinity on microbial composition and structure is still a research hotspot at present; however, most studies have only focused on this, and the correlation between microorganisms and CO_2_ emissions in salinized soils has been inadequately studied, especially for soil fungi. The Yellow River Delta (YRD), a land regressive and transgressive area, has formed soil with different salinization degrees due to different distances from the sea, land forming times, and soil desalination degrees. The soil salinity shows a tendency to decrease from sea to land and increase outward from both sides of the river [21]. It is a natural laboratory for studying the relationship between the saline soil microbial community and the salinity gradient [22]. In recent years, studies on soil microorganisms in the YRD have mainly concerned the structure and diversity of the bacterial community in saline or oil-contaminated soils [23], the effects of environmental factors on the bacterial community [24], and the relationship between the bacterial community and halophytic vegetation succession [25]; however, research on the changes in the structure of soil fungal communities under different salinity gradients in this area is almost non-existent. Microorganisms in the natural environment do not exist as independent individuals [26]; interactions among microbial species have a strong influence on their community stability [27], and the importance of network interactions for ecosystem processes and functions may exceed species diversity. In this work, we used high-throughput sequencing technology to explore the characteristics of the structure of soil fungal communities under salinity gradients in the Yellow River Delta and then combined this with the Partial Least Squares–Path Model to reveal whether the fungal communities influence CO_2_ emissions. We sought to provide a theoretical microbial perspective for future restoration efforts in wetland environments.

The purposes of this study were (1) to investigate the effects of salinity on the fungal community structure and fungal community interactions and (2) to investigate whether fungal community diversity significantly affects CO_2_ emissions.

## 2. Materials and Methods

### 2.1. Study Site

The YRD, located on the south bank of Bohai Bay and the west bank of Laizhou Bay (118°44”14.1” E–118°55”10.3” E, 37°26”16.7” N–37°32”41.4”), has a warm temperate, semi-humid continental monsoon climate with four distinct seasons. The annual average temperature is 12.9 °C, annual average rainfall is 596 mm, and annual evaporation is 1900–2400 mm. The main soil types in the YRD are fluvo-aquic soil and saline soil, and the soil parent material is Yellow River alluvium. The vegetation types are few, and the structure is simple. There are halophytes, such as *Suaeda salsa* (L.) *Pall.*, *Aeluropus sinensis* (D.) *Tzvel.*, *Tamarix Chinensis Lour.*, *Imperata cylindrica* (L.) *Beauv.*, and *Artemisiacapillaris Thunb*, with different salt tolerance levels.

### 2.2. Soil Sampling

The changes in soil salinity and soil type in the Yangtze River Delta were fully considered through several surveys. In October 2018, soil samples were collected in the Yellow River Delta at a distance of no less than 1 km between every two sampling points, as shown in Figure 1. The 30 soil samples were divided into three levels according to the salinity gradient (10 sampling points per level): high-salinity, medium-salinity, and low-salinity. Considering the distance of each sampling point in the same level, each level was divided into two groups: high-salinity (H1 and H2), medium-salinity (M1 and M2), and low-salinity (L1 and L2). In other words, a total of 30 soil samples were collected, with sampling points within each group (5 sampling points) that were relatively close to each other; the sampling points between groups were relatively far apart. The surface vegetation and cover were removed, and 0–20 cm of the soil was collected using a soil auger according to the diagonal five-point sampling method. After removing the gravel and roots, we placed the soil into two sealed bags (approximately 200 g each). One part was placed in an icebox, brought back to the laboratory within 24 h, and frozen at −80 °C until it was used for molecular biology research; the other part of the fresh sample was processed according to experimental requirements to determine the soil physicochemical properties and to perform laboratory incubation experiments.

### 2.3. Physicochemical Properties of the Soil Samples

Soil electrical conductivity (EC) was measured using an electrical conductivity meter (DDS-307A, China) with a soil–water suspension (soil/water ratio = 1:5). Because EC has always been regarded as a representative index of soil total soluble salt, the EC was used to characterize the soil salinity [28]. The sample soils were classified into three categories according to their salinity status: high-salinity soils (EC1:5 > 3 dS·m^−1^), medium-salinity soils (1.5 dS·m^−1^ < EC1:5 < 3 dS·m^−1^), and low-salinity soils (EC1:5 < 1.5 dS·m^−1^). Soil texture was measured using a laser particle size analyzer (Mastersizer 3000, Britain). The soil total nitrogen (TN) and organic matter (SOM) were determined using a macro element analyzer (Vario MACRO Cube, Germany). The alkali solution diffusion method was used to determine the soil available nitrogen (AN). The Olsen method was used to determine the soil available phosphorus (AP). Soil moisture content (MC) was determined by a drying-weighing method at 105 °C. The rate of soil CO_2_ emissions was evaluated using a laboratory incubation method. Fresh soil equivalent to 60 g of dry soil was weighed into a 250 mL culture bottle and pre-incubated at 25 °C for 7 days to activate the soil microorganisms. Taking into account the frequent inundation of the coastal areas of the Yellow River Delta by seawater and the evaporation of soil moisture in the incubator, the soil water–soil ratio was adjusted to 1:1 with sterile water during the formal incubation process to approach the original water content; this was followed by an unsealed incubation at 25 °C in the dark for a total of 14 days. During the culture period, the water in the bottle was supplemented by a weighing method, and the CO_2_ concentration was measured on day 1, 2, 3, 4, 7, 10, 13, and 14. The bottle mouth was sealed with a flip stopper when collecting gas; the gas was pushed into the bottle three times with a 10 mL syringe to ensure that the gas was evenly mixed, a 5 mL gas sample was drawn, and its concentration was measured with an Agilent gas chromatograph and then extracted with a syringe. We added 5 mL of air to the culture bottle, kept the pressure in the bottle consistent, and after 40 min, measured the concentration of the gas sample again; we then used the concentration difference between the two measurements to calculate the CO_2_ emission rate on that day. The CO_2_ emission rate for each site was calculated by averaging the CO_2_ emission rates for 8 days. The CO_2_ emission rate was calculated as
$$F=\frac{∆C\times V\times 44\times 273÷(273+T)}{t\times M\times 22.4}$$
where F’s unit is in μg kg^−1^-d, ΔC is the difference between the two CO_2_ concentrations, V is the volume of the incubation flask, T is the incubation temperature, t is the incubation time, and M is the sample weight [29].

### 2.4. DNA Extraction, PCR Amplification, and High-Throughput Sequencing

The total DNA was extracted from 0.5 g of soil per sample using a Fast DNA Kit (MP Biomedicals, Santa Ana, CA, USA). The detailed steps were carried out according to the instructions of the kit. The primers 528F (GCGGTAATTCCAGCTCCAA) and 706R (AATCCRAGAATTTCACCTCT) were used to amplify the V4 region of the 18S rRNA genes. The PCR reaction system included 10 ng of Genomic DNA, 5.0 μL of 10 × PCR Buffer, 1 μL of 50 μM each primer, 0.5 μL of 10 μM dNTPs, and 0.5 μL of 5 U/μL Plantium Taq DNA; this was replenished to 50 μL with sterilized, double-distilled water. The PCR reaction conditions were as follows: pre-denaturation for 30 s (94 °C), 30 cycles of denaturation for 20 s (94 °C), annealing for 20 s (60 °C), and extension for 20 s (72 °C), followed by a final extension for 5 min (72 °C) and preservation at 10 °C. The PCR products were purified and recovered by a Gene JET Kit (Thermo Scientific, Waltham, MA, USA) and were subjected to high-throughput sequencing on a Thermofisher Ion S5TMXL titanium platform. Cutadapt software (V1.9.1) was used to remove the low-quality reads and to trim the barcode and primer sequences [30]. The operational taxonomy units (OTUs) were clustered at a 97% identity threshold using the UPARSE algorithm (v7.0.1001) (Edgar, 2013). The UCHIME algorithm (v4.2.40) was used to detect and remove the chimerisms [31]. The RDP classifier (v2.0) was used to compare the OTU sequences with the SILVA132 database, with 80% similarity, to classify the sequences at the phylum, class, order, family, genus, and species levels [32].

### 2.5. Statistical Analysis

In order to compare soil physicochemical properties and fungal community alpha diversity between different saline gradient classes, we performed a one-way analysis of variance (ANOVA) using SPSS (v20.0) software [21]. In order to investigate Pearson’s correlation between the soil salinity and the alpha diversity of fungal communities, we used SPSS software. In order to determine the influence of soil physical and chemical factors on the fungal community structure, we performed a redundancy analysis (RDA) using CANOCO 5.0 software. In order to compare the differences between the salinity gradient classes in the structure of soil fungal communities, we performed a UPGMA clustering analysis using R software [33]. In order to analyze the contribution of the major fungal genera to the community differences, we performed a SIMPER analysis and ANOSIM test using PAST (V1.0) software [34]. Molecular ecological networks of soil fungi with high, medium, and low salinity gradients were constructed using online tools available on the MEAN website (http://ieg2.ou.Edu/mena/ (accessed on 13 January 2022)) [35]. Gephi software (V0.9.2) was used to visualize the networks [36]. Models were constructed using the “plspm” package in the R language, and goodness-of-fit statistics were used as a predictive power for the path models [37].

## 3. Results

### 3.1. Physical and Chemical Properties of the Soil Samples

The physical and chemical property data of soil samples are represented as the means ± SE, as shown in Table 1. According to the texture classification standard established by the United States Department of Agriculture (USDA), the soil in these six sites belonged to silt loam. The soil salinity ranged from 0.28 to 4.65 dS·m^−1^, and according to the level of soil salinity at the sampling points, the soils were classified with a high- (H1 or H2), medium- (M1 or M2), or low-salinity (L1 or L2) gradient. TN, SOM, and AN were the highest at the M2 site and the lowest at the H1 site, while AP was the highest at the M2 site and the lowest at the L2 site. However, there were no significant differences in the soil physicochemical properties between the two groups of the same salinity class, suggesting that distance has little effect on the soil environment within the YRD. These results show that there were significant differences in soil physical and chemical factors along the salinity gradient, forming a certain ecological gradient. The highest soil SOM content was found in the medium-salinity soil, the second highest was found in the low-salinity soil, and the lowest was found in the high-salinity soil.

### 3.2. Fungal Community Alpha Diversity

The fungal community’s coverage in the six sites was greater than 96%, indicating that the sequencing depth can reasonably represent the situation of the samples (Table 2). The number of fungal OTU increased as the salinity gradient decreased. The Shannon index of the fungal community at the H2 site was the largest, so the fungal diversity in the H2 site was the highest, followed by the H1, L1, L2, M2, and M1 sites. Abundances of fungal communities under different salinity gradients are ordered from highest to lowest as follows: L2, L1, M2, M1, H2, and H1. Taken together, the number of OTUs and abundance of the soil fungal communities increased as the soil salinity gradient decreased, and there was no significant difference in the distance on the alpha diversity in the same salinity class. In addition, a Pearson’s correlation test between soil salinity and fungal α-diversity showed that the number of OTUs, Chao1 index, and ACE index had the highest correlation coefficient with soil salinity, i.e., −0.66, 0.61, and −0.60, respectively; this indicates that soil salinity was the dominant factor affecting the number of OTUs, Chao1 index, and ACE index of the fungal communities, whereas the soil salinity had an insignificant effect on the Shannon index of fungal communities.

### 3.3. Fungal Community Structure

A total of 192 fungal genera belonging to eight phyla were identified by high-throughput sequencing in the YRD. The soil fungal communities mainly included Ascomycota, Basidiomycota, Mucoromycota, and Chytridiomycota at the phylum level (Figure 2a). Ascomycota was widely distributed in various sites, and their relative abundance was 54.81–74.65%, which was the dominant fungal phylum in the YRD. The Mucoromycota relative abundance was 4.90–25.74%, which was the subdominant fungal phylum in the YRD. The relative abundance of the top 30 fungal genera is shown in Figure 2b. *Chaetomium* was the dominant fungal genus at L1 and L2, with a relative abundance of 19.95% and 25.56%, respectively. *Alternaria* (13.68%), *Cephaliophora* (16.97%), *Fusarium* (6.56%), and *Alternaria* (9.46%) were dominant at H1, H2, M1, and M2, respectively. Taken together, the fungal community composition was similar under the same salinity class and varied more under different salinity classes, indicating that soil salinity has a greater effect on the fungal community structure than distance in the YRD.

### 3.4. Differences in the Structure of Fungal Community

The analyses of the ANOSIM test and UPGMA clustering based on the Bray–Curtis distance value showed that the soil fungal communities had significantly different distribution patterns under different salinity gradients (r = 0.504, *p* < 0.01, Figure 2). The UPGMA clustering analysis showed that the fungal communities under the same salinity gradient were grouped into one cluster, while the fungal communities under different salinity gradients were scattered. This indicates that soil salinity has a greater influence on soil fungal communities than geographic distance, and fungal community structure was quite different under different salinity gradients in the YRD. A SIMPER analysis was used to further find the different species that lead to different distribution patterns of fungal communities. Table 3 listed the contribution rates of the major fungal genera to spatial dissimilarity. The difference in the fungal community structure between the H and M groups mainly came from *Fusarium*, *Chaetomium*, and *Alternaria*, and the total contribution rate was 31.39%; *Chaetomium*, *Mortierella*, and *Malassezia* greatly contributed to the difference in the fungal community structure between the H and L groups, with a total contribution rate of 63.85%; the difference in the fungal community structure between the M and L groups was mainly due to *Chaetomium*, *Mortierella*, and *Fusarium*.

### 3.5. Effects of Environmental Factors on Fungal Community

For further insight into the relationship between fungal communities and soil environmental factors under different salinity gradients, a redundancy analysis (RDA) was conducted. As shown in Figure 3, RDA axes 1 and 2 explained 86.74% of the total variation. The correlation between fungal genera and environmental factors was expressed by the cosine of the angle between them, and the longer the arrow of environmental factors, the greater the influence on the fungal community structure. The results of the RDA forward selection showed that EC, T, AP, AN, TN, and clay caused significant differences in the fungal community distribution patterns under different salinity gradients (F = 6.6, *p* < 0.01), and Monte Carlo permutations tests indicated that all six factors were significantly correlated with the soil fungal community structure (*p* < 0.05). The long arrows of EC, AN, and AP factors show that they had a great influence on the fungal community distribution, and the explanation rate of EC was the highest, which was the dominant factor affecting the fungal community distribution patterns. TN, AN, and clay were positively correlated with most fungal genera, while EC, T, and AP were negatively correlated with most fungal genera; this indicates that TN, AN, and clay played a promotive role in the growth of most fungi, while EC, T, and AP played an inhibitory role in the growth of most fungi.

### 3.6. Molecular Ecological Network of Fungal Community

To investigate the interactions between the soil fungal OTUs under different salinity gradients and to determine the key species in the soil fungal communities, fungal OTUs that appeared in more than 50% of the samples were selected to construct molecular ecological networks (Table 4; Figure 4). Among them, each node signified a fungal OTU, the node size represented a node degree, different colors of the nodes represented different modules, and lines between nodes were colored based on the modules. In addition, 100 random networks with the same number of nodes and connections as the original networks were constructed by the random method, and their topological characteristics were calculated; in contrast, the cohesion of empirical networks was generally higher than that of random networks, which indicated that the interactions between fungal OTUs in the molecular ecological networks were significant. The decrease in salinity changed the network structure of fungal communities and increased the complexity of the networks. According to Table 4 and Figure 5, the node quantity, edge quantity, and modularity coefficients of the networks increased as the salinity gradient decreased. In high-salinity soil, there were 12 network modules with 2–14 nodes, and the modularity coefficient was 0.63 (Table 4, Figure 5a); in medium-salinity soil, there were 10 network modules with 2–28 nodes, and the modularity coefficient was 0.73 (Table 4, Figure 5b); in low-salinity soil, there were 19 network modules with 2–25 nodes, and the modularity coefficient was 0.76 (Table 4, Figure 5c). The average degree (Avg K) and average clustering coefficient (Avg CC) values of the medium-salinity network were the highest, while the average path distance (GD) was the shortest. Therefore, in medium-salinity soil, the information, energy, and material among fungal OTUs had the highest transmission efficiency and closest connection. At the same time, the response speed of microorganisms was the fastest, and the microbial community was more prone to change in medium-salinity soil. In this study, OTU_107 (Ascomycota; *Prussia*), OTU_52 (Ascomycota; *Talaromyces*), and OTU_515,778 (Ascomycota; *Aspergillus*, Ascomycota; *Scopulariopsis*) from Ascomycota had the highest number of connections, which were the core nodes of the molecular ecological network of high-, medium-, and low-salinity soil fungi, respectively. The relative abundance of Ascomycota was the highest in the soil fungal community in the YRD, indicating that Ascomycota occupied an important position in the saline soil environment and played a key role in maintaining the stability of the fungal community.

### 3.7. Impact of CO_2_ Emissions and Soil Fungal Community

To investigate the effect of fungi on CO_2_ emissions, a partial least squares regression analysis of the CO_2_ emission rate of soils in the study area was performed with the soil fungal and soil environmental factors (Figure 6a). It was found that the CO_2_ emission significantly increased with decreasing soil salinity, and the effect of geographical distance on soil CO_2_ emission rate was not significant. The PLS-PM goodness-of-fit was 0.6223, which proved that the PLS-PM was reasonably constructed and statistically significant (Figure 6b). The results showed that the fungal diversity had an effect on the CO_2_ flux (estimate: 1.08, *p* < 0.05). Salinity was significantly negatively correlated with the amount of fungal diversity (estimate: −0.58, *p* < 0.05), while AP, AN, TN, and SOM were significantly positively correlated with the fungal diversity (estimate: 0.41, *p* < 0.05; estimate: 1.55, *p* < 0.05; estimate: 0.58, *p* < 0.05; estimate: 0.40, *p* < 0.05). This confirms that soil environmental factors indirectly affect CO_2_ emissions by influencing fungal communities, and fungal communities significantly affect CO_2_ emissions.

## 4. Discussion

### 4.1. Differences in the Structures of Fungal Community under Different Salinity Gradients

Ascomycota was the predominant phylum with the highest abundance in the present study, which was similar to the results obtained by Wang et al. (2020) using molecular biology methods to study fungal samples in a saline environment [27]. The vast majority of Ascomycota fungi are saprophytes, which can decompose refractory organic substances such as lignin and keratin and play an important role in nutrient cycling [38]. Mucoromycota had the highest abundance in low-salinity soil, but Cryptomycota had the highest abundance in high-salinity soil. This suggests that Cryptomycota is more salt-tolerant than Mucoromycota. With the decrease in salinity in the soil, plant growth and species [39], plant rhizosphere exudates [40], and other factors could cause changes in the microbial composition and structure, which may be the reason for the decreased distribution of *Chaetomium* in high-salinity soil. Salinity had a significant effect on microbial diversity [9]. The results of the diversity analysis of fungal communities showed that the number of OTUs and abundance of the soil fungal communities increased as the soil salinity gradient decreased. Yang et al. highlighted in their study of soil fungi in the YRD that significantly lower values of the Chao1 richness index were observed in extreme salinity soil [21], which was also confirmed in the present study. The Pearson’s correlation test showed that the number of OTUs, Chao1 index, and ACE index were significantly negatively correlated with soil salinity, which could be attributed to the increase in the extracellular osmolarity of fungi caused by the accumulation of salt in the soil and that the fungi that were not adapted to osmotic stress were inhibited or even died [41], thus reducing the number of fungal OTUs, Chao1 index, and ACE index. Consistent with the findings of Yang et al., salinity altered the fungal community structure [21]. The UPGMA cluster analysis showed that there were obvious similarities in the structures of fungal communities with the same salinity gradient, but there were great differences in the structures of fungal communities under different salinity gradients, which could be due to the similar soil environments and similar effects on fungal communities under the same salinity gradient, but also due to different soil environments under different salinity gradients, so that the effects on fungal communities were different. The RDA analysis indicated that EC, T, AP, AN, TN, and clay had a significant effect on the fungal community structure. EC had the greatest influence, which was the main factor leading to the difference in the distribution patterns of fungal communities under different salinity gradients. Chowdhury et al. (2011) found that soil salinity affected the composition of soil microbial communities through osmotic potential, and fungi were more sensitive to salinity than bacteria [42]. Rajaniemi and Allison (2009) demonstrated that the effect of soil salinity on the composition of soil microbial communities is greater than that of soil C and N, which is consistent with the conclusion of this paper [43]. The SIMPER analysis showed that *Chaetomium*, *Mortierella*, and *Fusarium* were the fungal groups with the highest contribution to the difference in community structure, with an average relative abundance of 5.32%, 2.31%, and 0.89%, respectively. Among these, *Mortierella*, which was significantly correlated with EC, T, and AN (*p* < 0.05), could degrade the toxic organic compounds in soil, prevent soil degradation, and improve soil health [44]. *Fusarium* was significantly correlated with EC, TN, AN, and clay (*p* < 0.01). *Chaetomium*, part of the Ascomycota phylum, was significantly correlated with EC, T, AN, TN, and AP (*p* < 0.05); it could produce a large number of cellulolytic enzymes and plays an important role in the carbon cycle of the natural ecosystem and soil improvement.

### 4.2. Molecular Ecological Networks of Fungal Community under Different Salinity Gradients

In the natural environment, microorganisms often form complex network structures through various interactions rather than as independent individuals [45]. The positive correlations between microorganisms may mean that there are positive ecological interactions, such as commensalism or mutualism [46]. The negative correlations may be attributed to competition or amensalism [47]. Zheng et al. found that all six networks had positive association percentages above 98% in their study of soil microbial responses to salt stress in Bohai Bay, China, which is similar to the results of the present study [20]. In the three fungal networks with different salinity in this study, the percentages of the positive connections among fungal OTUs were all above 90%, which could be because the high-salinity habitat forced the fungi to strengthen their cooperation in response to salt stress, or because the fungi in the high-salinity environment had mutualism in the long-term co-evolution process. The module is a closely connected area in the network, which is usually interpreted as a niche [48]. The low-salinity network had the largest number of modules (19) and the highest modularity (0.76); therefore, the niche differentiation degree of microorganisms in the low-salinity soil was the highest, and the community structure was the most complex. The nodes with the highest number of connections in the network were identified as the core nodes [49]. The absence of the core nodes may cause module and network decomposition [50], so they play an important role in maintaining the stability of the microbial community. Core nodes were usually interpreted as key species [51]. OTU_107, OTU_52, OTU_515, and OTU_778 from Ascomycota had the most connections, which were identified as the core nodes of the fungal molecular ecological networks. Ascomycota also had the highest relative abundance in the fungal community in the YRD, which indicated that Ascomycota occupied an important position in the saline soil environment and played a key role in maintaining the stability of the fungal community.

### 4.3. Impact of CO_2_ Emissions and Soil Fungal Community

Soil properties influence C cycling by altering wetland microbial diversity, and this is an important but previously underestimated indirect pathway [52]. Soil CO_2_ emission is an important indicator that responds to the participation of soil microorganisms in the carbon cycle process and converts organic matter [53]. Soil fungi not only release CO_2_ during the metabolic decomposition of organic matter but also participate in the carbon sequestration processes to reduce CO_2_ emissions [54]. It was found that increased soil salinity indirectly reduces CO_2_ emissions by reducing soil fungal diversity. This is mainly because increased salinity has a strong negative effect on fungal community activity. For example, elevated salinity in the soil increases the extracellular osmotic pressure rate of fungi, which inhibits or even kills fungal activity and ultimately leads to a decrease in fungal diversity [28]. Increases in soil organic matter and TN increase soil CO_2_ emissions because soil with higher organic matter and TN content tend to have higher soil C and N content, resulting in strong soil respiration and high CO_2_ emissions [55]. Moreover, the decomposition process of organic matter by saprophytic fungi releases CO_2_ [56], and a study by Suvendu et al. found a significant positive correlation between the saprophytic fungus *Mortierella* and CO_2_ emissions [57]. In the present study, *Mortierella* was one of the genera that contributed most to the differences in soil fungal community structure at different salinities and was significantly correlated with EC. This suggests that salinity can influence CO_2_ emissions by affecting fungal communities. Therefore, it can be inferred that CO_2_ emissions from the Yellow River Delta are closely related to the existence of soil fungal communities, while soil environmental factors mainly affect soil CO_2_ emissions indirectly by influencing the fungal communities.

## 5. Conclusions

Our study found that the soil fungal abundance increased as the soil salinity decreased. EC had the greatest, most significant impact on the fungal community structure, which was the dominant factor leading to the difference in the distribution patterns of the fungal communities under different salinity gradients. *Chaetomium* was the dominant fungal genus in the low-salinity soil, while *Aspergillus* was the dominant fungal genus in the high- and medium-salinity soil. The SIMPER analysis showed that *Chaetomium*, *Fusarium*, *Mortierella*, *Alternaria*, and *Malassezia* were the dominant fungal groups leading to the difference in the structures of fungal communities under different salinity gradients. In the molecular ecological networks, the decrease in salinity changed the reticulation of fungal communities and increased the complexity of the network. Moreover, fungal community diversity affects CO_2_ emissions, soil environmental factors also affect CO_2_ emissions by influencing fungal communities, and increased soil salinity decreases soil CO_2_ emissions.

## Figures and Tables

**Figure 1 ijerph-20-04190-f001:**
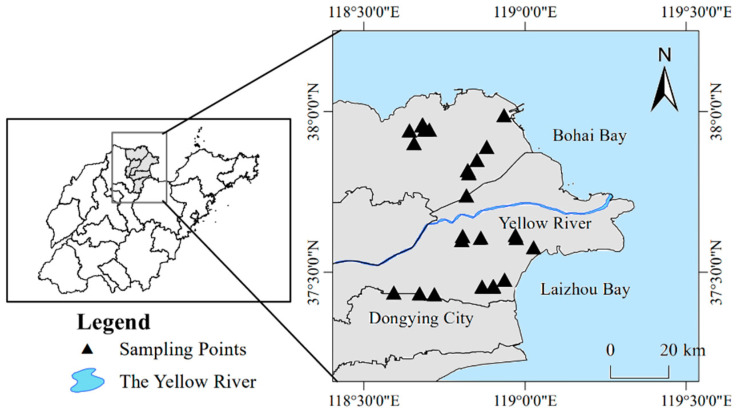
Yellow River Delta sampling sites.

**Figure 2 ijerph-20-04190-f002:**
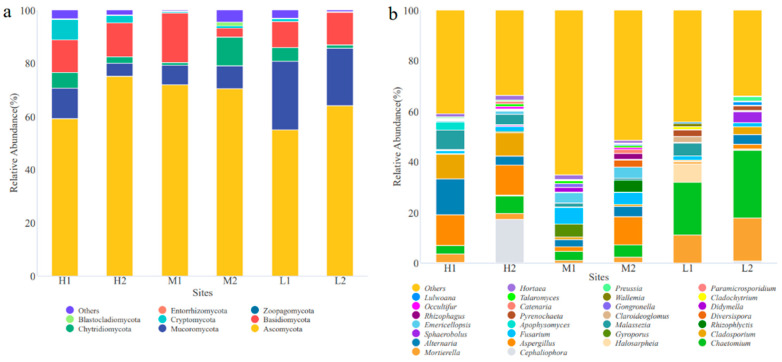
Fungal community structural composition at different sampling sites. (**a**) At the phylum level; (**b**) at the genus level.

**Figure 3 ijerph-20-04190-f003:**
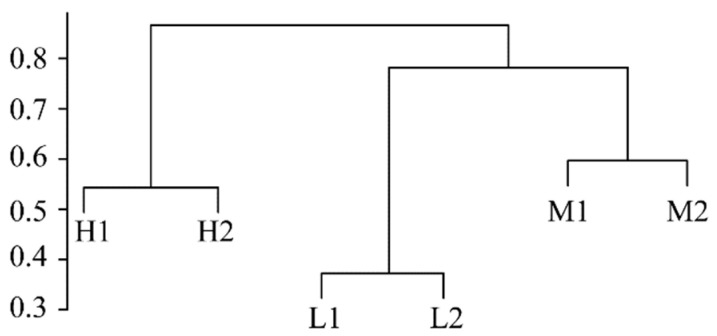
UPGMA clustering analysis based on the Bray–Curtis distance.

**Figure 4 ijerph-20-04190-f004:**
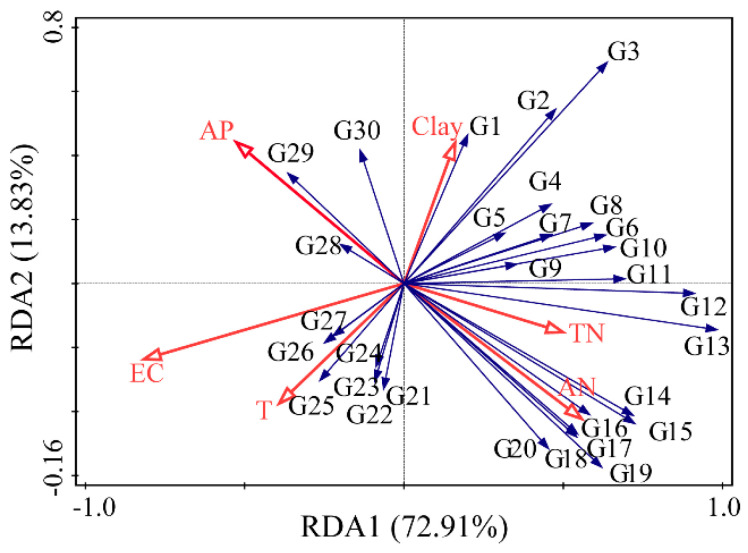
RDA analysis with dominant fungal genera and environmental factors in the Yellow River Delta. G1, *Aspergillus*; G2, *Psathyrella*; G3, *Mortierella*; G4, *Plectosphaerella*; G5, *Spizellomyces*; G6, *Sphaerobolus*; G7, *Preussia*; G8, *Alternaria*; G9, *Olpidium*; G10, *Phymatotrichopsis*; G11, *Pyrenochaeta*; G12, *Pulvinula*; G13, *Chaetomium*; G14, *Cladosporium*; G15, *Halosarpheia*; G16, *Tetracladium*; G17, *Gongronella*; G18, *Rhizoctonia*; G19, *Fusarium*; G20, *Mucor*; G21, *Cephaliophora*; G22, *Rhizophlyctis*; G23, *Emericellopsis*; G24, *Trichoderma*; G25, *Tilletiopsis*; G26, *Gyroporus*; G27, *Talaromyces*; G28, *Hortaea*; G29, *Gongronella*; G30, *Malassezia*.

**Figure 5 ijerph-20-04190-f005:**
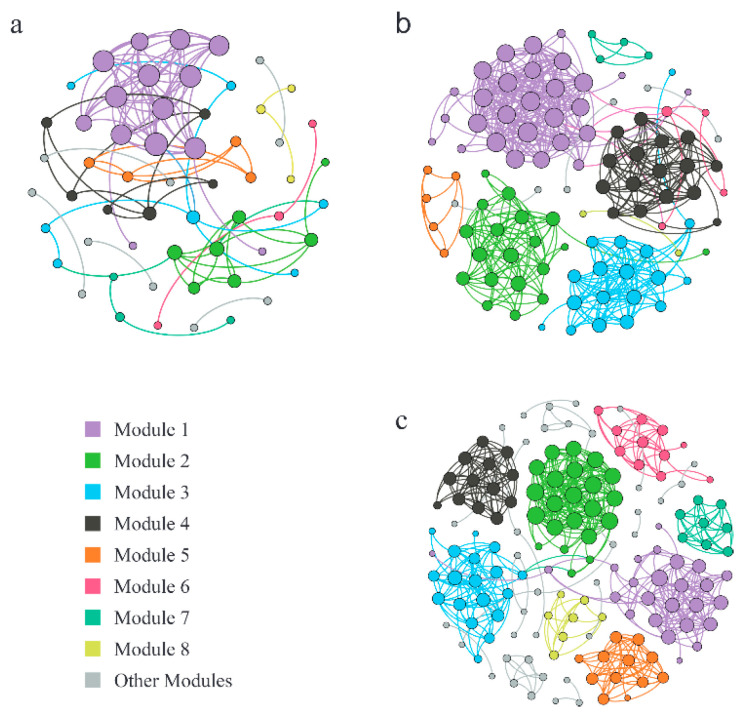
Molecular ecological networks of the fungal communities in different salinity gradients. (**a**) high-salinity gradient; (**b**) medium-salinity gradient; (**c**) low-salinity gradient.

**Figure 6 ijerph-20-04190-f006:**
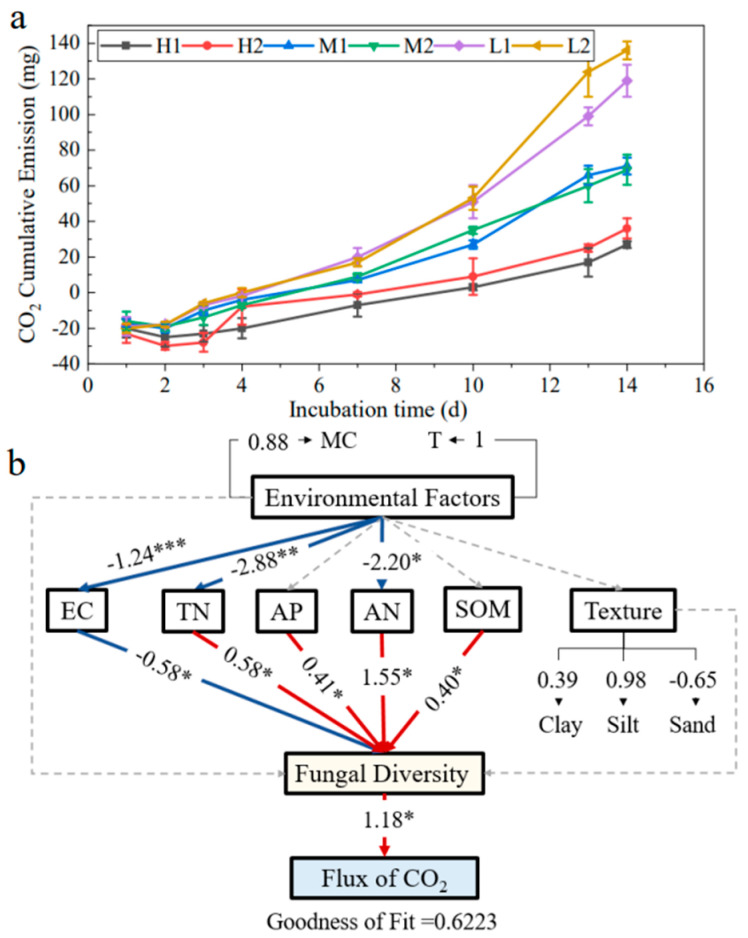
Soil CO_2_ cumulative emission (**a**) and Partial Least Squares–Path Model (PLS-PM) of the effect of the soil non-microbiological indicators and microbial indicators on CO_2_ fluxes (**b**). Coefficients significantly different from 0 are indicated by * *p* ≤ 0.05, ** *p* ≤ 0.01, and *** *p* ≤ 0.001.

**Table 1 ijerph-20-04190-t001:** Soil physicochemical properties at different sampling sites.

Groups of	H		M		L	
Sites	H1	H2	M1	M2	L1	L2
T (°C)	18.26 ± 1.55 a	19.24 ± 0.64 a	14.96 ± 0.59 bc	16.88 ± 1.53 ab	13.54 ± 2.81 c	12.56 ± 1.59 c
MC (%)	18.33 ± 2.27 ab	20.62 ± 2.14 a	19.57 ± 2.45 ab	19.35 ± 0.91 ab	15.94 ± 3.43 bc	14.58 ± 4.09 c
EC (dS·m^−1^)	4.65 ± 0.88 a	4.61 ± 1.05 a	2.38 ± 0.62 b	2.61 ± 0.48 b	0.28 ± 0.16 c	0.43 ± 0.31 c
TN (g·Kg^−1^)	0.22 ± 0.06 b	0.25 ± 0.07 b	0.38 ± 0.18 ab	0.53 ± 0.10 a	0.50 ± 0.21 a	0.33 ± 0.12 ab
SOM (g·Kg^−1^)	4.39 ± 0.38 e	6.29 ± 0.40 d	8.24 ± 0.37 b	9.30 ± 0.42 a	7.26 ± 0.40 c	7.28 ± 0.39 c
AP (mg·Kg^−1^)	2.84 ± 0.06 b	2.52 ± 0.03 bc	2.54 ± 0.07 b	3.53 ± 0.08 a	2.47 ± 0.08 bc	2.16 ± 0.54 c
AN (mg·Kg^−1^)	16.61 ± 0.70 e	18.62 ± 0.44 e	29.28 ± 0.48 d	41.17 ± 1.34 a	36.31 ± 0.44 b	31.92 ± 0.23 c
Sand (%)	21.57 ± 0.30 b	24.34 ± 0.40 a	21.36 ± 0.25 b	17.34 ± 0.56 d	20.57 ± 0.79 c	21.42 ± 0.47 b
Silt (%)	75.62 ± 0.32 bc	73.06 ± 0.37 d	75.25 ± 0.30 c	79.10 ± 0.45 a	76.03 ± 0.767 b	75.27 ± 0.45 c
Clay (%)	2.81 ± 0.03 c	2.60 ± 0.04 d	3.39 ± 0.05 b	3.56 ± 0.14 a	3.40 ± 0.03 b	3.31 ± 0.05 b

Notes: Different letters in a single line indicate significant (*p* < 0.05) differences among the soil samples. H stands for high-salinity soil; M stands for medium-salinity soil; L stands for low-salinity soil, the same below. Sand (%) refers to the particle content in the soil with a particle size of 0.05–2 mm; Silt (%) refers to the particle content in the soil with a particle size of 0.002–0.05 mm; Clay (%) refers to the particle content in the soil with a particle size of less than 0.002 mm.

**Table 2 ijerph-20-04190-t002:** Fungal community alpha diversity indices in different sampling sites.

Group	H		M		L	
Sites	H1	H2	M1	M2	L1	L2
OTU	114 ± 27.00 c	120 ± 8.94 c	143 ± 25.38 bc	180 ± 66.52 ab	187 ± 37.88 ab	233 ± 35.27 a
Shannon	3.00 ± 0.05 ab	3.09 ± 0.06 a	2.7 ± 0.08 c	2.78 ± 0.09 c	2.97 ± 0.09 b	2.95 ± 0.09 b
ACE	166.42 ± 27.56 c	190.19 ± 15.21 bc	202.62 ± 28.90 bc	222.31 ± 55.20 ab	238.37 ± 34.78 ab	268.02 ± 34.67 a
Chao1	161.41 ± 31.20 c	186.51 ± 9.28 bc	196.21 ± 32.65 bc	229.14 ± 54.5 ab	235.08 ± 30.42 ab	272.65 ± 39.09 a
Coverage (%)	97.74 ± 1.22 bc	96.08 ± 2.36 c	98.76 ± 1.18 ab	99.60 ± 0.38 ab	99.76 ± 0.26 a	99.85 ± 0.14 a

Notes: Different letters in a single line indicate significant (*p* < 0.05).

**Table 3 ijerph-20-04190-t003:** Contribution rates of the major fungal genera to community dissimilarity in the SIMPER analysis.

Genera	Average Abundance (%)			
	H	M	Contribution (%)	Cumulative (%)
*Fusarium*	0.03	1.27	12.72	12.72
*Chaetomium*	0.11	1.43	9.44	22.16
*Alternaria*	0.25	0.78	9.23	31.39
	**H**	**L**	**Contribution (%)**	**Cumulative (%)**
*Chaetomium*	0.11	14.40	38.53	38.53
*Mortierella*	0.07	6.53	20.30	58.83
*Malassezia*	0.18	0.74	5.02	63.85
	**M**	**L**	**Contribution (%)**	**Cumulative (%)**
*Chaetomium*	1.43	14.40	34.41	34.41
*Mortierella*	0.33	6.53	17.05	51.46
*Fusarium*	1.27	1.35	5.44	56.90

**Table 4 ijerph-20-04190-t004:** Topological properties of the soil fungal community molecular ecological networks under different salinity gradients and the related properties of random networks.

	Network Indices	Groups		
		H	M	L
0 Molecular ecological networks	Cutoff	0.89	0.89	0.89
	Nodes	57	102	152
	Edges	112	511	590
	Avg K	3.93	10.02	7.76
	Avg CC	0.43	0.76	0.67
	GD	2.94	1.64	2.45
	Modularity	0.63	0.73	0.76
	Modules	12	10	19
Random networks	Avg CC	0.12 ± 0.024	0.15 ± 0.012	0.08 ± 0.009
	GD	2.93 ± 0.110	2.38 ± 0.031	2.78 ± 0.032

Note: Avg K, average degree; Avg CC, average clustering coefficient; GD, average path distance.

## Data Availability

The data will be available upon request.

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
