# Peer review of "Soil Fungal Community Structure and Its Effect on CO2 Emissions in the Yellow River Delta"

_ijerph, 2023, doi:10.3390/ijerph20054190_

Round 1

Reviewer 1 Report (Previous Reviewer 3)

The manuscript entitled “Soil fungal communities structure and its effect on CO2 emissions in the Yellow River Delta” is very interesting work based on the different salt soil content. The authors studied the physicochemical properties of the soil sample and the fungal communities’ structure by high-throughput sequencing. First, they explored the physicochemical properties of the soil samples and calculated the CO2 emission rate. In this paper was showed how the fungi influence CO2 emissions under salinity gradients – low, medium and high. The obtained results were then combined with the molecular ecological networks to find the adaptation mechanisms of fungi to salt stress. Fungal species diversity is lowest in the high saline soil and greatest in the low saline soil with the most representatives from Ascomycota, Basidiomycota and Mucuromycota. The difference in salinity gradient had not only a significant effect on fungal communities' structure and diversity, but and on all analyzed indexes, such as operational taxonomy units number, Chao1 and ACE indexes of the fungal communities, but no significant effect on the Shannon index. In general, an increase in fungal community abundance increases CO2 emissions.

However, this version of the manuscript is much more understandable as the authors have gone to great lengths and introduced additional explanations. My only comments are that the Conclusions should be written as text, not listed with numbers and the number of Keywords should be reduced to five. it is not good to use abbreviations in the abstract, especially since they are introduced in the text.

Author Response

Reviewer 2 Report (New Reviewer)

Soil carbon cycle has become a global concern, and soil salinization affects the structural characteristics of soil fungi and thus affects the related carbon cycle. The topic of this research is very clear, but there are still some problems that need to be modified to improve the readability of the paper. Specific suggestions are as follows:

Abstract:

1. Relevant data should be added in the abstract to further support results, for example, the changes of fungal communities under different salinity gradients;

2. The scientific significance of this study was not clear;

3. Abbreviations that appear for the first time should have a full description;

4. Description of data analysis methods were not necessary in the abstract.

Introduction:

1. The first paragraph and the second paragraph of the introduction should be merged, and appropriate connection between the two paragraphs should be added, so as to highlight the interaction between fungi and organic carbon release;

2. Although there is a lack of relevant studies in the Yellow River Delta, there are still a lot of relevant studies in high-salinity areas in other regions, s as to microbial changes under salt stress. Please add relevant information to better introduce the research focus of this study.

3. Simplify the purposes to 2.

Method:

1. Supplement the number of soil samples collected and the number of repetitions

Results:

1. Move the relevant discussion to the section 4.

Discussion:

1. Generally speaking, the discussion part is insufficient. It is suggested to add relevant further explanations to enhance the explanation of the conclusion;

2. It is suggested to merge 4.1 and 4.2;

3. Increase the microbial regulation mechanism of co2 emission in 4.4, for example, which microbial species will be affected first under different salinity conditions, so as to influence co2 emission.

The conclusion:

1. It is recommended to use a single paragraph to describe the conclusion of the paper, rather than presenting it in strips.

Soil carbon cycle has become a global concern, and soil salinization affects the structural characteristics of soil fungi and thus affects the related carbon cycle. The topic of this research is very clear, but there are still some problems that need to be modified to improve the readability of the paper. Specific suggestions are as follows:

Abstract:

1. It is suggested to add relevant data support in the abstract to further support the changes of fungal communities under different salinity gradients;

2. It is suggested to add the scientific significance of this study at the end of the abstract;

The introduction part;

1. It is suggested that the first paragraph and the second paragraph of the introduction should be merged, and appropriate connection between the two paragraphs should be added, so as to highlight the interaction between fungi and organic carbon release;

2. Although there is a lack of relevant studies in the Yellow River Delta, there are still a lot of relevant studies in high-salinity areas in other regions. Therefore, the introduction should add the progress of relevant studies in other regions and the change characteristics of microbial salt stress, so as to better introduce the research focus of this study.

3. There are 4 purposes mentioned in the research purposes, and it is suggested to simplify them to 2

Experimental method part:

1. Supplement the number of soil samples collected and the number of repetitions

Analysis part:

1. Analysis is often an elaboration of results, so it is suggested to move the relevant references in 3.6 Analysis to the discussion section.

Discussion section:

1. Generally speaking, the discussion part is insufficient. It is suggested to add relevant further explanations to enhance the explanation of the conclusion;

2. It is suggested to merge 4.1 and 4.2;

3. The discussion about microbial regulation mechanism of CO2 emission should be added in section 4.4, for example, which microbial species will be affected first under different salinity conditions, so as to influence CO2 emission.

Conclusion:

1. It is recommended to use a single paragraph to describe the conclusion of the paper, rather than presenting it in strips.

Author Response

Reviewer 3 Report (New Reviewer)

The manuscript " Soil fungal community structure and its effect on CO2 emissions in the Yellow River Delta" is a very interesting work based on soils with different salinity gradients. The authors investigated the structure of the fungal community and CO2 emissions from soil samples by combining high-throughput sequencing and laboratory incubation method. In this paper, the authors show the community structure of fungi at different salinities and how fungi affect CO2 emissions. The manuscript is well structured. The Conclusions drawn correspond to the stated results. But there are still a few small problems. Below is a detailed list of suggested edits and questions.

1. Check the Keywords. "fungal communities" should be " fungal community". 

2. Change "fungal communities" to "fungal community" in the title of the chart, on line 225. same comment for other chart titles.

3. Lines 35-36: more recent references should be used in this paper.

4. Check the English in line 379 and line 403.

Author Response

Reviewer 4 Report (New Reviewer)

The work is exciting and done correctly. New techniques and ways to present the genus and groups of fungi identified in soils of varying salinity found in the Yellow River Delta in China. The processing of the obtained results is interesting and quite extensive. The conclusion based on CO2 emission by different groups of fungi at varying salinity is very interesting. It brings new knowledge on the impact of salinity on the formation of biodiversity of fungal communities and the usefulness of OTU in presenting this relationship. Kudos to the authors for their patience in their research and imaginations in presenting results. 

Author Response

     Thank you for your letter about our manuscript ‘Soil fungal communities structure and its effect on CO2 emissions in the Yellow River Delta’ (ijerph-2220325). We wish you good luck in your research.

Reviewer 5 Report (New Reviewer)

The paper explores the characteristics of the fungal microbiome in the soil at varying salinity gradients in the Yellow River Delta.

Overall comment: The paper is well presented and fits well within the scope of this journal. I believe it will attract a lot of readership from this journal.

Suggestion/s: Is it possible for the authors to conduct a brief comparative study to correlate their results with existing data, if possible?

Author Response

Thank you for your letter about our manuscript ‘Soil fungal communities structure and its effect on CO2 emissions in the Yellow River Delta’ (ijerph-2220325). Those comments are all valuable and very helpful for revising and improving our paper, as well as the important guiding significance to our research.

We added a brief comparative study to relate their results to the available data.

In lines 372-374: Yang et al. highlighted in their study of soil fungi in YRD that significantly lower values of the Chao1 richness index were observed in extreme salinity soil, which was also confirmed in the present study.

In lines 379-380: Consistent with the findings of Yang et al. salinity altered the fungal community structure.

Round 2

Reviewer 2 Report (New Reviewer)

I believe the authors have modified the manuscript carefully according to the comments.

This manuscript is a resubmission of an earlier submission. The following is a list of the peer review reports and author responses from that submission.

Round 1

Reviewer 1 Report

Overall, I found this to be an interesting study.  However, I found some problems with the presentation of the work, including: lack of clarity in the purpose/importance of the study; inappropriate references throughout the introduction; not referencing very relevant (almost identical) work (Yang and Sun 2020, Front. Microbiol. doi.org/10.3389/fmicb.2020.594284); lack of clarity in methods, especially in statistical methods; some statements in results that seem unfounded; and some aspects of the study that don't seem to contribute to the overall story (i.e. the CO2 respiration work).  Below is a detailed list of suggested edits and questions.

Line 33-34: Maybe inappropriate reference.  Did this study show how much land was suffering from salinity?  Looking at the title it seems like a study on how salinity affected AMF.

Line 35: Same comment for reference 2

Line 37: Same comment for reference 3.   A reference for the overall effects of salinity on soil properties would more appropriately come from a source like Weil and Brady’s textbook Nature and Properties of Soils, instead of a study that investigated salinity effects on only soil microbes.  Perhaps you mean to cite a reference found within the introduction of that paper?

Lines 37-40: Sentence is unclear.  Which parameter is used to assess soil quality, organic matter decomposition, nutrient use, nutrient cycling?  Also, is the reference cited here one example of a study that used one of these parameters to assess soil quality?  Because it seems like a more specific study and seems like a more appropriate reference could be cited.

Lines 47-48: Reference 4 is inappropriate here.  The cited study is about N acquisition and cycling and not about CO2 fluxes.  I suggest checking all references since I think I will not have time to check each one.

Lines 51-54: You have written, “Plants form symbiotic relationships with mycorrhizal 51 fungi that firmly anchor carbon in the soil, and scientists have learned that a special type 52 of mycorrhizal fungus called ectomycorrhizal fungi is helping plants to take up carbon 53 dioxide more quickly [11].

and cited: “11. Rath, K.M.; Fierer, N.; Murphy, D.V.; Rousk, J. Linking bacterial community composition to soil salinity along environ-510 mental gradients. Isme Journal 201913, 836-846, doi:10.1038/s41396-018-0313-8. “  Please give an appropriate reference.  I am going to assume there was an error with an automated reference generator and not highlight these anymore. 

Line 54: take up carbon more quickly because the mycorrhizal fungi are increasing plant growth or photosynthesis?  Also, your study site mentions vegetation that is mostly arbuscular mycorrhizal so why mention ectomycorrhizae at all?  Especially since AMF has been shown to reduce salt stress in plant hosts.

Line 57: “under CO2 conentrations” Do you mean “increasing atmospheric CO2 concentrations?”

Line 61-62: A map or diagram could be a good addition in the methods section if this gradient was used in the study.

Lines 66-69: It would be good to clarify that the bacterial communities have been studied but not the fungal communities, since fungi are also microbes.

Line 69: “The interaction between soil fungi”  I’m not sure what you mean.  Perhaps it would be good to introduce the “what” and “why study this” of fungal interactions, otherwise the reader is left wondering what fungal interactions you are talking about and why they are important.

Lines 70-72: Can one really look at the mechanisms by which fungi adapt to salt stress via observing changes to community composition?  I would think you could only look to see if some fungi were salt tolerant and others not, or speculate that certain taxa found across the gradient were either tolerant or adapted as soils became more saline with salt water intrusion.  

Line 73: Rational utilization and development have not yet been introduced and could use a couple of sentences somewhere early in the introduction so the reader knows what is at stake here.

Line 92: “six sampling sites” Is that six sites per soil salinity class, or six total, and if six total, how many at each class?

Lines 112-125: The method used to measure CO2 or soil respiration.  Did it use the -80C frozen soil or the air dried soil, and wouldn’t using either (instead of fresh or in place) greatly skew the results, especially if you are interested in the CO2 flux due to fungal activity, since the fungal communitiy now consists only of the fungi that were able to quickly grow in rehydrated soil (from spores, hyphal fragments, etc.)?  Also, using DI water would change which fungi did the decomposing and the rate at which that occurred, no?

Lines 129-130: Were there adapters for sequencing libraries?

Line 134: Only 5 cycles of annealing and extension for library prep? 

Lines 146-148: Purpose of statistical methods isn’t totally clear.  Please restate the goal/intent for each step.  E.g. “In order to compare soil physicochemical properties among different saline gradient classes, we used ANOVA…” “In order to test…we used Pearson’s correlation method.”  Was ANOVA used to compare each individual response variable between saline classes?  If so, one could argue that MANOVA would be more appropriate as a first test here to show sites are different in soil parameters since you are looking at many response variables.   Similarly for the community analysis, can you please write the statistical methods in a way that the reader can clearly understand the purpose of each statistical approach or test?

Line 162: typo with superscript “1”

Line 162-163: “salinity gradient change was formed in the region” is unclear.  Do you mean that your sampling revealed a low-med-high gradient in the region?

Lines 165-167: I’m not sure these soil property results show a soil salinity gradient or any kind of ecological gradient since the most nutrient-rich soils were form the medium salinity samples and the poorest soils, in terms of nutrients and SOM were from the high and low salinity areas (N and P, respectively).  There may be a salinity gradient among the sample sites, but these other soil factors could be seen as additional factors that may also affect fungal communities, in addition to salt.  This would require a model-based approach.

Lines 167-168: By what measure are you comparing “soil nutrient content?” TN?  AP?  SOM?  Looking at the table, it seems like this is only true for SOM.  Can this be explained by vegetation?

Lines 172-177: notes for a table should be in smaller font, but I assume this will be changed for the final version.

Line 179: “fungal coverage” is confusing, I suggest rewording.

Lines 187-191: The Pearson’s correlation test is not clear.  What were the factors included in the test?  Did it include all other soil parameters?  And why were there two measures of diversity used (Shannon and Chao), but only one included in the analysis?  Also, is there an explanation for the greater OTU richness but decreased evenness (evidenced by lower Shannon index) in the lower salinity plots?

Lines 201-203: awkward wording, please rephrase.

Figure 2: I don’t understand the relative abundance measure on the y-axis of the graphs.  What are these abundances relative to?  I would expect each bar to go to 100% with percentage of each Phylum and Genus to occupy a relative portion of the bar, but maybe I’m missing something.  Do these graphs simultaneously show total fungal abundance as well as community composition?  If so, please explain.

Lines 240-241: I don’t think wording, “different spatial distribution patterns” is correct here, but I could be wrong.  I think what you mean is that the ANOSIM showed that fungal communtites were different.

Lines 319-322: I am not an expert on network models but the claims in these lines seem pretty big.  Is it possible to draw conclusions about fungal “response speed” and such with this information?

Lines 340-342: Incomplete sentence, meaning is unclear.

Section 4.4: This section seems to be stating the obvious, that more salinity leads to lower microbial activity and less respiration.  It is not clear what the importance of this aspect of the study is.

Section 5: The conclusion reads more like a summary or an abstract.  What is the main finding, or the most important contribution?

Author Response

Response to Reviewer #1’s comments

Thank you for your letter about our manuscript ‘Soil fungal communities structure and its effect on CO2 emissions in the Yellow River Delta’ (ijerph-2067058). Those comments are all valuable and very helpful for revising and improving our paper, as well as the important guiding significance to our research. We have studied the comments carefully and have made corrections which we hope meet with approval. Thank you again for your positive and constructive comments and suggestions on our manuscript. We hope you will find our revised manuscript acceptable for publication.

We have also edited the MDPI website in English based on expert advice.

Comment 1: Line 33-34: Maybe inappropriate reference. Did this study show how much land was suffering from salinity? Looking at the title it seems like a study on how salinity affected AMF.

Response: Thank you for your constructive comments. We have reselected the reference in line 33-34. We cited this paper because he did mention them in the text, and then we found the paper that mentioned them first and replaced them.

The new reference is: Rengasamy, P. World salinization with emphasis on Australia. Journal of Experimental Botany 2006, 57, 1017-1023, doi:10.1093/jxb/erj108.

Comment 2: Line 35: Same comment for reference 2

Response: Thank you for your advice. As with reference 1, we modify reference 2 to: Shrivastava, P.; Kumar, R. Soil salinity: A serious environmental issue and plant growth promoting bacteria as one of the tools for its alleviation. Saudi Journal of Biological Sciences 2015, 22, 123-131, doi:10.1016/j.sjbs.2014.12.001.

Comment 3: Line 37: Same comment for reference 3. A reference for the overall effects of salinity on soil properties would more appropriately come from a source like Weil and Brady’s textbook Nature and Properties of Soils, instead of a study that investigated salinity effects on only soil microbes. Perhaps you mean to cite a reference found within the introduction of that paper?

Response: Thank you for your constructive comments. As with reference 2, we modify reference 3 to: Nair, P.K.R. The Nature and Properties of Soils, 13th Edition.By N. C. Brady and R. R. Weil. Agroforestry Systems 2002, 54, 249-249, doi:10.1023/A:1016012810895.

Comment 4: Lines 37-40: Sentence is unclear.  Which parameter is used to assess soil quality, organic matter decomposition, nutrient use, nutrient cycling?  Also, is the reference cited here one example of a study that used one of these parameters to assess soil quality?  Because it seems like a more specific study and seems like a more appropriate reference could be cited.

Response: Thank you for your constructive comments. We have made a change in line 37-40 to text: In the soil ecosystem, microorganisms play a key role in organic matter decomposition [4] and nutrient cycling [5], and microbial community structure is often used as an important parameter to measure the change in soil quality [6].  

[4] Gmach, M.R.; Cherubin, M.R.; Kaiser, K.; Cerri, C.E.P. Processes that influence dissolved organic matter in the soil: a review. Scientia Agricola 2020, 77, e20180164-e20180164, doi:10.1590/1678-992x-2018-0164.

[5] Fellbaum, C.R.; Mensah, J.A.; Pfeffer, P.E.; Kiers, E.T.; Bucking, H. The role of carbon in fungal nutrient uptake and transport: implications for resource exchange in the arbuscular mycorrhizal symbiosis. Plant signaling & behavior 2012, 7, 1509-1512, doi:10.4161/psb.22015.

[6] van Leeuwen, J.P.; Djukic, I.; Bloem, J.; Lehtinen, T.; Hemerik, L.; de Ruiter, P.C.; Lair, G.J. Effects of land use on soil microbial biomass, activity and community structure at different soil depths in the Danube floodplain. European Journal of Soil Biology 2017, 79, 14-20, doi:10.1016/j.ejsobi.2017.02.001.

Comment 5: Lines 47-48: Reference 4 is inappropriate here.  The cited study is about N acquisition and cycling and not about CO2 fluxes.  I suggest checking all references since I think I will not have time to check each one.

Response: Thank you for your constructive comments. We have reselected the reference in lines 47-48. The revised reference is: Xin, Y.; Ji, L.; Wang, Z.; Li, K.; Xu, X.; Guo, D. Functional Diversity and CO2 Emission Characteristics of Soil Bacteria during the Succession of Halophyte Vegetation in the Yellow River Delta. International Journal of Environmental Research and Public Health 2022, 19, doi:10.3390/ijerph191912919.

Comment 6: and cited: “11. Rath, K.M.; Fierer, N.; Murphy, D.V.; Rousk, J. Linking bacterial community composition to soil salinity along environ-510 mental gradients. Isme Journal 2019, 13, 836-846, doi:10.1038/s41396-018-0313-8. “  Please give an appropriate reference.  I am going to assume there was an error with an automated reference generator and not highlight these anymore.

Response: Thank you for your constructive comments. We checked the references in the article. We have reselected the reference in lines 47-48 The revised reference is: Combarnous, Y.; Thi Mong Diep, N. Cell Communications among Microorganisms, Plants, and Animals: Origin, Evolution, and Interplays. International Journal of Molecular Sciences 2020, 21, doi:10.3390/ijms21218052.

Comment 7: Line 54: take up carbon more quickly because the mycorrhizal fungi are increasing plant growth or photosynthesis?  Also, your study site mentions vegetation that is mostly arbuscular mycorrhizal so why mention ectomycorrhizae at all?  Especially since AMF has been shown to reduce salt stress in plant hosts.

Response: Thank you for your constructive comments. Considering the content and contextual logic of the article, we modified lines 54-56 to: The relationship between soil fungal communities and CO2 emission fluxes needs to be dissected to reveal changes in the structure of soil fungal communities under increas-ing atmospheric CO2 concentrations.

Comment 8: Line 57: “under CO2 conentrations” Do you mean “increasing atmospheric CO2 concentrations?”

Response: Thank you for your constructive comments. Yes, we meant "the increase in atmospheric CO2 concentration" has been modified in the manuscript.

Comment 9: Line 61-62: A map or diagram could be a good addition in the methods section if this gradient was used in the study.

Response: Thank you for your constructive comments. The salinity gradient we use is based on the electrical conductivity of the soil sample. We have added the sampling point map.

Comment 10: Lines 66-69: It would be good to clarify that the bacterial communities have been studied but not the fungal communities, since fungi are also microbes.

Response: Thank you for your constructive comments. we modified lines 62-67 to: In recent years, studies on soil microorganisms in the YRD have mainly concerned the structure and diversity of the bacterial community in the saline or oil-contaminated soils [21], the effects of environmental factors on the bacterial community [22], and the relationship between the bacterial community and halophytic vegetation succession [23], but the research on the changes in the structure of soil fungal communities under different salinity gradients in this area is almost non-existent.

Comment 11: Line 69: “The interaction between soil fungi” I’m not sure what you mean.  Perhaps it would be good to introduce the “what” and “why study this” of fungal interactions, otherwise the reader is left wondering what fungal interactions you are talking about and why they are important.

Response: Thank you for your constructive comments. We added content in line 67-71: Microorganisms in the natural environment do not exist as independent individuals [24]. Interactions among microbial species have a strong influence on their community stability [25] and the importance of network interactions for ecosystem processes and functions may exceed species diversity.

Comment 12: Lines 70-72: Can one really look at the mechanisms by which fungi adapt to salt stress via observing changes to community composition?  I would think you could only look to see if some fungi were salt tolerant and others not, or speculate that certain taxa found across the gradient were either tolerant or adapted as soils became more saline with salt water intrusion. 

Response: Thank you for your constructive comments. We have rewritten this sentence in lines 72-75. In this work, we used high-throughput sequencing technology to explore the characteristics of the structure of soil fungal communities under salinity gradients in the Yellow River Delta and then combined this with the Partial Least Squares Path Model to reveal whether the fungal communities influence CO2 emissions.

Comment 13: Line 73: Rational utilization and development have not yet been introduced and could use a couple of sentences somewhere early in the introduction so the reader knows what is at stake here.

Response: Thank you for your constructive comments. We have rewritten this sentence in lines 75-76: We seek to provide a theoretical microbial perspective for future restoration efforts in wetland environments.

Comment 14: Line 92: “six sampling sites” Is that six sites per soil salinity class, or six total, and if six total, how many at each class?

Response: The 6 sampling points are: the soil salinity is divided into high, medium and low 3 levels, the same salinity level set 2 parallel, a total of 6 subgroups.

Comment 15 Lines 112-125: The method used to measure CO2 or soil respiration.  Did it use the -80C frozen soil or the air dried soil, and wouldn’t using either (instead of fresh or in place) greatly skew the results, especially if you are interested in the CO2 flux due to fungal activity, since the fungal communitiy now consists only of the fungi that were able to quickly grow in rehydrated soil (from spores, hyphal fragments, etc.)?  Also, using DI water would change which fungi did the decomposing and the rate at which that occurred, no?:

Response: Thank you for your comments. We use frozen soil at -80C. Pre-culture at 25C for 7 days to activate soil microorganisms to increase experimental reliability. Also the water added for pre-culture and formal culture is sterile water and the inaccurate description in the article has been corrected.

Comment 16: Lines 129-130: Were there adapters for sequencing libraries?

Response: Dear expert, our fungal sequencing was given to Beijing Novogene, we asked the company this question and it was The amplicon project contains adapter sequence adapters adapted to the sequencer, with respect to sequencing adapters.

 5' Adapter:

5'-AATGATACGGCGACCACCGAGATCTACAC(i5Index)

ACACTCTTTCCCTACACGACGACGCTCTTCCGATCT-3'

3' Adapter:

5'-CAAGCAGAAGACGGCATACGAGAT (reverse complementary sequence of i7Index) GTGACTGGAGTTCAGACGTGTGCTCTCTTCCGATC-3'.

Comment 17: Line 134: Only 5 cycles of annealing and extension for library prep?

Response: Thank you for your comments. We have checked and found it to be 30 cycles and have made changes to the text.

Comment 18: Lines 146-148: Purpose of statistical methods isn’t totally clear.  Please restate the goal/intent for each step.  E.g. “In order to compare soil physicochemical properties among different saline gradient classes, we used ANOVA…” “In order to test…we used Pearson’s correlation method.”  Was ANOVA used to compare each individual response variable between saline classes?  If so, one could argue that MANOVA would be more appropriate as a first test here to show sites are different in soil parameters since you are looking at many response variables.   Similarly for the community analysis, can you please write the statistical methods in a way that the reader can clearly understand the purpose of each statistical approach or test?

Response: Thank you for your comments. We rewrite this paragraph: In order to compare soil physicochemical properties and fungal community alpha diversity between different saline gradient classes, we performed a one-way analysis of variance (ANOVA) using SPSS (v20.0) software. In order to investigate Pearson’s cor-relation between soil salinity and the alpha diversity of fungal communities, we used SPSS software. In order to determine the influence of soil physical and chemical factors on the fungal community structure, we performed redundancy analysis (RDA) using CANOCO 5.0 software. In order to compare differences between salinity gradient classes in the structure of soil fungal communities, we performed UPGMA clustering analysis using R software [31]. In order to analyze the contribution of the major fungal genera to the community differences, we performed SIMPER analysis and ANOSIM test using PAST (V1.0) software [32]. Molecular ecological networks of soil fungi with high, medium, and low-salinity gradients were constructed using online tools available on the MEAN website (http://ieg2.ou.Edu/mena/) [33]. The Gephi (V0.9.2) software was used to visualize the networks [34]. Models were performed using the "plspm" package in the R language, and goodness-of-fit statistics were used as a predictive power for the path models [35].

Comment 19: Line 162: typo with superscript “1”

Response: Thank you for mentioning that detail, we have modified -1.

Comment 20: Line 162-163: “salinity gradient change was formed in the region” is unclear.  Do you mean that your sampling revealed a low-med-high gradient in the region?

Response: Thank you for your comments. We have rewritten this sentence in lines 173-175: The soil salinity ranged from 0.28 to 4.65 dS·m-1, and according to the level of soil salinity at the sampling points, soils were classified with a high- (H1 or H2), medium- (M1 or M2), or low-salinity (L1 or L2) gradient.

Comment 21: Lines 165-167: I’m not sure these soil property results show a soil salinity gradient or any kind of ecological gradient since the most nutrient-rich soils were form the medium salinity samples and the poorest soils, in terms of nutrients and SOM were from the high and low salinity areas (N and P, respectively).  There may be a salinity gradient among the sample sites, but these other soil factors could be seen as additional factors that may also affect fungal communities, in addition to salt.  This would require a model-based approach.

Response: Thank you for your comments. This study is supposed to ensure the consistency of physicochemical factors other than salinity at all sites. However, the soil environment at each site was not homogeneous during the large-scale sampling survey,  Therefore, at the end of the paper, we used the “plspm” model to investigate the effect of each physicochemical factor on the soil fungal community. Moreover, RDA analysis in the paper showed that EC, T, AP, AN, TN and Clay had significant effects on fungal community structure, with EC having the greatest effect and being the dominant factor leading to differences in fungal community distribution patterns under salinity gradients (P<0.05).

Comment 22: Lines 167-168: By what measure are you comparing “soil nutrient content?” TN?  AP?  SOM?  Looking at the table, it seems like this is only true for SOM.  Can this be explained by vegetation?

Response: Thank you for your comments. We have rewritten this sentence in lines 179-181: The highest soil SOM content was found in the medium-salinity soil, the second-highest was found in the low-salinity soil, and the lowest was found in the high-salinity soil. For the current study, it cannot be solved with vegetation for the time being, but thanks to the expert's reminder, we will add data on vegetation (e.g., dry weight) to solve this problem in the subsequent study.

Comment 23: Lines 172-177: notes for a table should be in smaller font, but I assume this will be changed for the final version.

Response: Thank you for your comments. We have reduced the font size of the table notes

Comment 24: Line 179: “fungal coverage” is confusing, I suggest rewording.:

Response: Thank you for your comments. We have reselected the fungal coverage in lines 191: fungal community's coverage.

Comment 25: Lines 187-191: The Pearson’s correlation test is not clear.  What were the factors included in the test?  Did it include all other soil parameters?  And why were there two measures of diversity used (Shannon and Chao), but only one included in the analysis?  Also, is there an explanation for the greater OTU richness but decreased evenness (evidenced by lower Shannon index) in the lower salinity plots?

Response: Thank you for your comments. We have rewritten this sentence in lines 199-205: In addition, Pearson's correlation test between soil salinity and fungal α-diversity showed that the number of OTUs, the Chao1 index, and the ACE index had the highest correlation coefficient with soil salinity, i.e., -0.66, 0.61, and -0.60, respectively, which indicated that soil salinity was the dominant factor affecting the number of OTUs, the Chao1 index, and the ACE index of the fungal communities, while soil salinity had an insignificant effect on the Shannon index of fungal communities. Also, The Shannon index was used to estimate one of the microbial diversity indices in the sample. Higher values indicate higher community diversity. However, the number of OTUs was Observed species. Therefore, it may occur that the situation in the text.

Comment 26: Lines 201-203: awkward wording, please rephrase. Figure 2: I don’t understand the relative abundance measure on the y-axis of the graphs.  What are these abundances relative to?  I would expect each bar to go to 100% with percentage of each Phylum and Genus to occupy a relative portion of the bar, but maybe I’m missing something.  Do these graphs simultaneously show total fungal abundance as well as community composition?  If so, please explain.

Reply. Thank you for your comment. We performed high-throughput sequencing with primers 528F/706R, not only for the structure of soil fungi, but also for other eukaryotes. So it appears that each bar is not 100%. This time we took out the soil fungi in high-throughput sequencing separately and did the minimum draw level and redid the relative abundance histogram.

The results of this section were also reanalyzed. We have rewritten this paragraph in 3.3 Fungal community structure. We rewrite this paragraph: A total of 192 fungal genera belonging to 8 phyla were identified by high-throughput sequencing in the YRD. The soil fungal communities mainly included Ascomycota, Basidiomycota, Mucoromycota, and Chytridiomycota at the phylum level (Figure 1a). Ascomycota was widely distributed in various sites, and their relative abundance was 54.81–74.65%, which was the dominant fungal phylum in the YRD. The Mucoromycota relative abundance was 4.90–25.74%, which was the subdominant fungal phylum in the YRD. The relative abundance of Basidiomycota was 3.32–18.63%. The relative abundance of the top 30 fungal genera is shown in Figure 1b. Chaetomium was the dominant fungal genus at L1 and L2, with a relative abundance of 19.95% and 25.56%, respectively. Alternaria (13.68%), Cephaliophora (16.97%), Fusarium (6.56%), and Alternaria (9.46%) were dominant at H1, H2, M1 and M2, respectively.

Figure 2. Fungal communities structural composition in different sampling sites. (a) at the phylum level; (b) at the genus level

Comment 27: Lines 240-241: I don’t think wording, “different spatial distribution patterns” is correct here, but I could be wrong.  I think what you mean is that the ANOSIM showed that fungal communtites were different.

Response: Thank you for your comments. Considering more accurate wording, we change“different spatial distribution patterns” in the 228 line to “different distribution patterns”.

Comment 28: Lines 319-322: I am not an expert on network models but the claims in these lines seem pretty big.  Is it possible to draw conclusions about fungal “response speed” and such with this information?

Response: Thank you for your comments. Lines 319-322 could respond to the fastest information exchange between fungal species and the fastest response of the environment in a medium salinity soil. For the“response speed” no supporting literature was found at the moment.

Comment 29: Lines 340-342: Incomplete sentence, meaning is unclear.

Response: Thank you for your comments. we modified lines 324-326 to: The results showed that the fungal OTUs had the greatest effect on the CO2 flux (estimate: 0.646, p < 0.05), followed by the fungal ACE index (estimate: 0.552, p < 0.05).

Comment 30: Section 4.4: This section seems to be stating the obvious, that more salinity leads to lower microbial activity and less respiration.  It is not clear what the importance of this aspect of the study is.

Response: Thank you for your comments. We have rewritten this paragraph in 4.4: Soil CO2 emission is an important indicator that responds to the participation of soil microorganisms in the carbon cycle process and converts organic matter [50]. Soil fungi not only release CO2 during the metabolic decomposition of organic matter but also participate in carbon sequestration processes to reduce CO2 emissions [51]. It was found that increased soil salinity indirectly reduces CO2 emissions by reducing soil fungal abundance. This is mainly because increasing salinity has a strong negative effect on fungal community activity. For example, elevated salinity in soil increases the extracellular osmotic pressure rate of fungi, which inhibits or even kills fungal activity and ultimately leads to a decrease in fungal diversity [26]. Increases in soil organic matter, fungal OTUs, and fungal diversity increase soil CO2 emissions, because soil with higher organic matter content tends to have higher soil C content, resulting in strong soil respiration and high CO2 emissions. Therefore, it can be inferred that CO2 emissions from the Yellow River Delta are closely related to the existence of soil fungal communities, while soil environmental factors mainly affect soil CO2 emissions indirectly by influencing fungal communities.

Comment 31: Section 5: The conclusion reads more like a summary or an abstract.  What is the main finding, or the most important contribution?

Response: Thank you for your comments. We have rewritten this paragraph Section 5:

(1)           The soil fungal abundance increased as the soil salinity decreased.

(2)           EC had the greatest, most significant impact on the fungal community structure, which was the dominant factor leading to the difference in the distribution patterns of fungal communities under different salinity gradients (p < 0.05).

(3)           Chaetomium was the dominant fungal genus in the low-salinity soil, while Aspergillus was the dominant fungal genus in the high- and medium-salinity soil. Chaetomium, Fusarium, Mortierella, Alternaria, and Malassezia were the dominant fungal groups leading to the difference in the structures of fungal communities under different salinity gradients.

(4)           The results of molecular ecological networks showed that the decrease in salinity changed the reticulation of fungal communities and increased the complexity of the network.

(5)           Soil environmental factors also affect CO2 emissions by influencing fungal communities, and increased soil salinity decreases soil CO2 emissions.

Reviewer 2 Report

Author needs to revise the manuscript.

-abstract needs more accurate writing then literature contents.

-in introduction authors can write more information

-community analysis sections could be more improved.

-discussion part needs revision

-conslusion section could also be revised.

Author Response

Response to Reviewer #2’s comments

Thank you for your letter about our manuscript ‘Soil fungal communities structure and its effect on CO2 emissions in the Yellow River Delta’ (ijerph-2067058). Those comments are all valuable and very helpful for revising and improving our paper, as well as the important guiding significance to our research. We have studied the comments carefully and have made corrections which we hope meet with approval. Thank you again for your positive and constructive comments and suggestions on our manuscript. We hope you will find our revised manuscript acceptable for publication.

We have also edited the MDPI website in English based on expert advice.

Comment 1: abstract needs more accurate writing then literature contents.

Response: Thank you for your comments. We modified the paper according to the expert's comments and had the entire paper corrected in English in the official MDPI touch-up service. We have rewritten the abstract:

Abstract: Soil salinization is one of the most compelling environmental problems on a global scale. Fungi play a crucial role in promoting plant growth, enhancing salt tolerance, and inducing disease resistance. Moreover, microorganisms decompose organic matter to release carbon dioxide, and soil fungi also use plant carbon as a nutrient and participate in the soil carbon cycle. Therefore, we used high-throughput sequencing technology to explore the characteristics of the structures of soil fungal communities under different salinity gradients and whether the fungal communities influence CO2 emissions in the Yellow River Delta and then combined this with molecular ecological networks to reveal the mechanisms by which fungi adapt to salt stress. In the Yellow River Delta, a total of 192 fungal genera belonging to 8 phyla were identified, with Ascomycota and dominating in the fungal community. Soil salinity was the dominant factor affecting the number of OTUs, the Chao1 index, and the ACE index of the fungal communities, with correlation coefficients of -0.66, 0.61, and -0.60, respectively (p < 0.05). Salinity gradients promoted substantial differentiation in the fungal community structure. SIMPER analysis showed that Chaetomium, Fusarium, Mortierella, Alternaria, and Malassezia were the dominant fungal groups, leading to the differences in the structures of fungal communities under different salinity gradients. EC, T, AP, AN, TN, and Clay had a significant impact on the fungal community structure. EC had the greatest influence and was the dominant factor leading to the difference in the distribution patterns of fungal communities under different salinity gradients (p < 0.05). Fungal molecular ecological network analyses indicated that the node quantity, edge quantity, and modularity coefficients of the networks increased with the salinity gradient. The Ascomycota occupied an important position in the saline soil environment and played a key role in maintaining the stability of the fungal community. PLS-PM analysis showed that the number of OTUs and the ACE index in the fungal community had a significant positive effect on CO2 emissions.

Comment 2: in introduction authors can write more information.

Response: Thank you for your constructive comments. We have revised and added to the introduction.

Lines 35-39: Fungi are the main members of soil microorganisms and are widely distributed in terrestrial ecosystems. In the soil ecosystem, fungi play a key role in organic matter decomposition [4] and nutrient cycling [5], and the fungal community structure is often used as an important parameter to measure the change in soil quality [6].

Lines 62-71: In recent years, studies on soil microorganisms in the YRD have mainly concerned the structure and diversity of the bacterial community in the saline or oil-contaminated soils [21], the effects of environmental factors on the bacterial community [22], and the relationship between the bacterial community and halophytic vegetation succession [23], but the research on the changes in the structure of soil fungal communities under different salinity gradients in this area is almost non-existent. Microorganisms in the natural environment do not exist as independent individuals [24]. Interactions among microbial species have a strong influence on their community stability [25] and the importance of network interactions for ecosystem processes and functions may exceed species diversity.

Lines 72-76: In this work, we used high-throughput sequencing technology to explore the characteristics of the structure of soil fungal communities under salinity gradients in the Yellow River Delta and then combined this with the Partial Least Squares Path Model to reveal whether the fungal communities influence CO2 emissions. We seek to provide a theoretical microbial perspective for future restoration efforts in wetland environments.

Comment 3: community analysis sections could be more improved.

Response: Thank you for your constructive comments. We have revised and added to the community analysis sections.

Lines 199-200: In addition, Pearson's correlation test between soil salinity and fungal α-diversity showed that……

Lines 202-212: A total of 192 fungal genera belonging to 8 phyla were identified by high-throughput sequencing in the YRD. The soil fungal communities mainly included Ascomycota, Basidiomycota, Mucoromycota, and Chytridiomycota at the phylum level (Figure 1a). Ascomycota was widely distributed in various sites, and their relative abundance was 54.81–74.65%, which was the dominant fungal phylum in the YRD. The Mucoromycota relative abundance was 4.90–25.74%, which was the subdominant fungal phylum in the YRD. The relative abundance of Basidiomycota was 3.32–18.63%. The relative abundance of the top 30 fungal genera is shown in Figure 1b. Chaetomium was the dominant fungal genus at L1 and L2, with a relative abundance of 19.95% and 25.56%, respectively. Alternaria (13.68%), Cephaliophora (16.97%), Fusarium (6.56%), and Alternaria (9.46%) were dominant at H1, H2, M1 and M2, respectively.

Lines 227-229: The analyses of the ANOSIM test and UPGMA clustering based on the Bray–Curtis distance showed that the soil fungal communities had significantly different distribution patterns under different salinity gradients.

Comment 4: discussion part needs revision

Response: Thank you for your constructive comments. We have revised and added to the discussion.

Lines 335-342: Ascomycota was the predominant phylum with the highest abundance in the present study, which was similar to the results obtained by Wang et al. (2020) using molecular biology methods to study fungal samples in a saline environment [25]. The vast majority of Ascomycota fungi are saprophytes, which can decompose refractory organic substances such as lignin and keratin, playing an important role in nutrient cycling [39]. Mucoromycota had the highest abundance in low-salinity soil, but Cryptomycota had the highest abundance in high-salinity soil. This suggests that Cryptomycota is more salt-tolerant than Mucoromycota.

Lines 304-417: Soil CO2 emission is an important indicator that responds to the participation of soil microorganisms in the carbon cycle process and converts organic matter [50]. Soil fungi not only release CO2 during the metabolic decomposition of organic matter but also participate in carbon sequestration processes to reduce CO2 emissions [51]. It was found that increased soil salinity indirectly reduces CO2 emissions by reducing soil fungal abundance. This is mainly because increasing salinity has a strong negative effect on fungal community activity. For example, elevated salinity in soil increases the extracellular osmotic pressure rate of fungi, which inhibits or even kills fungal activity and ultimately leads to a decrease in fungal diversity [26]. Increases in soil organic matter, fungal OTUs, and fungal diversity increase soil CO2 emissions, because soil with higher organic matter content tends to have higher soil C content, resulting in strong soil respiration and high CO2 emissions. Therefore, it can be inferred that CO2 emissions from the Yellow River Delta are closely related to the existence of soil fungal communities, while soil environmental factors mainly affect soil CO2 emissions indirectly by influencing fungal communities.

Comment 5: conclusion section could also be revised.

Response: Thank you for your constructive comments. We have revised and added to the conclusion.

(1)           The soil fungal abundance increased as the soil salinity decreased.

(2)           EC had the greatest, most significant impact on the fungal community structure, which was the dominant factor leading to the difference in the distribution patterns of fungal communities under different salinity gradients (p < 0.05).

(3)           Chaetomium was the dominant fungal genus in the low-salinity soil, while Aspergillus was the dominant fungal genus in the high- and medium-salinity soil. Chaetomium, Fusarium, Mortierella, Alternaria, and Malassezia were the dominant fungal groups leading to the difference in the structures of fungal communities under different salinity gradients.

(4)           The results of molecular ecological networks showed that the decrease in salinity changed the reticulation of fungal communities and increased the complexity of the network.

(5)           Soil environmental factors also affect CO2 emissions by influencing fungal communities, and increased soil salinity decreases soil CO2 emissions.

Reviewer 3 Report

The manuscript entitled “Soil fungal communities structure and its effect on CO2 emissions in the Yellow River Delta” is very interesting work based on the different salt soil content. The authors studied the physicochemical properties of the soil sample and the fungal communities’ structure by high-throughput sequencing. First, they explored the physicochemical properties of the soil samples and calculated the CO2 emission rate. In this paper was showed how the fungi influence CO2 emissions under salinity gradients – low, medium and high. The obtained results were then combined with the molecular ecological networks to find the adaptation mechanisms of fungi to salt stress. Fungal species diversity is lowest in the high saline soil and greatest in the low saline soil with the most representatives from Ascomycota, Basidiomycota and Mucuromycota. The difference in salinity gradient had not only a significant effect on fungal communities' structure and diversity, but and on all analyzed indexes, such as operational taxonomy units number, Chao1 and ACE indexes of the fungal communities, but no significant effect on the Shannon index. In general, an increase in fungal community abundance increases CO2 emissions. The manuscript is well structured. The Conclusions drawn correspond to the stated results. The Discussion is very well written. My only remarks are directed to Figure 2 (it is not clear), Figire 4 (I can’t understand the modules on the Figure) and Figure 5 (too big). Please give the originals to Editor! I advise you to mark the exact sampling points on google maps! Congratulations! Your work is really impressive!

Author Response

Response to Reviewer #3’s comments

Thank you for your letter about our manuscript ‘Soil fungal communities structure and its effect on CO2 emissions in the Yellow River Delta’ (ijerph-2067058). Those comments are all valuable and very helpful for revising and improving our paper, as well as the important guiding significance to our research. We have studied the comments carefully and have made corrections which we hope meet with approval. Thank you again for your positive and constructive comments and suggestions on our manuscript. We hope you will find our revised manuscript acceptable for publication.

We have also edited the MDPI website in English based on expert advice.

Thank you for your constructive comments. We gave the originals of Figure 2 Figure 4 and Figure 5 to the editor and added the sampling point map.

Round 2

Reviewer 1 Report

Dear Authors,

Thank you for thoroughly addressing my comments from the original review.  I have several major concerns still.  

1. The CO2/respiration aspect of the study is troubling and I think it should be removed.  According to the methods section and your previous response, it appears that you 1) collected soil samples, 2) froze the samples at -80C, 3) incubated samples at 25C, flooded the samples with equal parts (1:1) fresh sterile water, and 4) measured CO2 emissions.  If you want to measure differences in respiration as it relates to fungal communities in these soils, these methods are troubling.  Passing soil through a sieve would first damage many fungi and alter the community.  Freezing at -80C would then kill many fungi, further altering the community.  Incubating soils saturated with fresh water would then tell you about respiration in a flooded (but not saline) environment.  Finally, you give no information on how you actually measured CO2 emission in the lab.  Assuming you actually did use some sort of closed-system analyzer, what you would be measuring is likely not at all representative of conditions in the field nor representative of the organisms that would have been respiring in the field.  Without much more detail about the methods and thorough, well-referenced evidence of why these methods are appropriate for your question, the entire CO2 aspect of this study should be removed.  You could still speculate on potential effects of the effects of salinity on respiration in the discussion by citing other relevant literature.

2. You do not cite Yang and Sun (2020; Frontiers in Microbiology, https://doi.org/10.3389/fmicb.2020.594284).  Their study is almost identical to yours and was completed in the same geographic region, with similar methods, similar statistical approaches (including network analysis), etc.  This is oddly suspicious that you would not reference the single most highly relevant study from the same region when discussing your results.  Please explain.

3.  Please make clear in the study design that you have 5 replicate samples from each of the 6 categories of soils.  Also, please explain why you kept 6 categories (e.g. H1 and H2) instead of combining to have three categories and 10 reps for each.  What is it about M1 and M2 that makes them different?

4. For the PLS-PM analysis, you included three measures of diversity, ACE, Shannon H', and OTU richness, which would be collinear in the model.  You should either show that they are not (which is highly unlikely) or you should choose one measure of diversity that makes the most sense for your question.

5. A reference is needed for the statement on lines 412-413.
